# A Hybrid State/Disturbance Observer-Based Feedback Control of Robot with Multiple Constraints

**DOI:** 10.3390/s22239112

**Published:** 2022-11-24

**Authors:** Du Xu, Tete Hu, Ying Ma, Xin Shu

**Affiliations:** 1School of Mechanical and Electrical Engineering, Central South University, Changsha 410083, China; 2Yonker Environmental Protection Co., Ltd., Changsha 410330, China

**Keywords:** manipulator, input and output constraints, trajectory tracking control, hybrid observer

## Abstract

Controlling the manipulator is a big challenge due to its hysteresis, deadzone, saturation, and the disturbances of actuators. This study proposes a hybrid state/disturbance observer-based multiple-constraint control mechanism to address this difficulty. It first proposes a hybrid state/disturbance observer to simultaneously estimate the unmeasurable states and external disturbances. Based on this, a barrier Lyapunov function is proposed and implemented to handle output saturation constraints, and a back-stepping control method is developed to provide sufficient control performance under multiple constraints. Furthermore, the stability of the proposed controller is analyzed and proved. Finally, simulations and experiments are carried out on a 2-DOF and 6-DOF robot, respectively. The results show that the proposed control method can effectively achieve the desired control performance. Compared with several commonly used control methods and intelligent control methods, the proposed method shows superiority. Experiments on a 6-DOF robot verify that the proposed method has good tracking performance for all joints and does not violate constraints.

## 1. Introduction

Robots have been widely used on many occasions and have worked in single-task environments and in complex tasks [1]. For example, the manipulator needs to not only grasp, carry, and stack, but also to cooperate with other machines to finish complex operations. This manipulator usually has strongly nonlinear features, such as hysteresis, input deadzone, output saturation, and a nonlinear load, as well as many unmeasurable physical states and disturbances, including friction, clearance, noise, etc. Ignoring these constraints may lead to an undesirable performance, such as a high steady-state error, poor transient response, and large overshoot [2]. In addition, a violation of constraints during operation may result in performance degradation, hazards, or system damage. Therefore, it is necessary to consider the input and output constraints of the robot system at the same time.

Many control methods have been designed for the input deadzone control, and most of this research is based on the deadzone function with assumed parameters. In general, the parameter of a deadzone function is difficult to obtain, which leads to common controllers is difficult to implement. There has been several studies conducted which aim to solve the problem of nonlinear systems with input deadzone [3]. For example, Wang et al. proposed a control method for a special nonlinear system with symmetrical deadzone [4] in which the parameters of deadzone function are unknown and the neural network is used to solve the deadzone effect. In paper [5], an adaptive neural network controller was proposed for single-master–multiple-slaves teleoperation with consideration to time delays and input deadzone uncertainties. Wu et al. [6] proposed a study on the observer-based controller for nonlinear systems with unmodeled dynamics and actuator deadzone. Tong et al. [7] employed neural networks to approximate the unknown nonlinear uncertainties. Hu et al. [8] presented an integrated direct/indirect adaptive robust control mehtod for a class of nonlinear systems, preceded by unknown nonsymmetric, nonequal slope deadzone nonlinearity. Sang et al. [9] presented a new adaptive iterative learning control approach with deadzone to address the problem of bounded noise. Wang et al. [10] proposed a nonlinear controller to overcome deadzone nonlinearities, which are unavoidable in many physical systems due to the imperfections of system components. The proposed control employs an ideal linear model of the system and a model controller to generate an ideal reference output. Although these methods have obtained many successful applications, they are basically designed for one or two kinds of nonlinearity and they do not consider controller design problems in the case of multiple nonlinear constraints. In addition, these methods require all physical states to be measurable and consider less external uncertainty, leading to poor control performance on many occasions.

To handle output saturation constraints, many techniques have been developed [11,12]. Yan et al. [13] introduced a nonlinear smooth function to tackle the saturation input nonlinearity, with a disturbance observer compensating for the unknown time-varying disturbances. In [14], an adaptive fuzzy output-feedback control was developed for a class of output-constrained and uncertain nonlinear systems with input saturation and unmeasured states using a log-type BLF and an auxiliary system incorporating the virtual control variable. Bu et al. [15] proposed a novel DDC algorithm, constructed using saturated output data, and output saturation causes the convergence rate of DDC systems to slow down. Xing et al. [16] proposed an observer-based adaptive control for uncertain nonlinear systems with input saturation and output constraints. Han et al. [17] proposed a dynamic parallel distributed compensator to design a dynamic output-feedback controller to ensure the finite time boundary and dissipate the singular uncertain time-varying-delay fuzzy systems, subject to actuator saturation and output constraints.

In recent years, the potential barrier Lyapunov function has attracted more and more attention and has become an effective tool for the control design of constrained nonlinear uncertain systems. The control law is derived directly from Lyapunov stability analysis. It has been successfully applied in the tracking control of a robotic manipulator [18] and in handling the full-state constraint control of a nonlinear system [19]. Thus, there are many types of BLF that have been proposed to cope with constraints, such as the log-type BLF [19], integral-type BLF [20], and tan-type barrier Lyapunov function (BLF) [19]. However, at the beginning of control, BLF-based controllers usually require specified time-varying constraints to achieve effective and stable control [18]. In addition, when the constraint variable is infinitely close to the predefined boundary, in order to meet the constraint conditions, the output of the BLF-based controller tends to be infinite, which leads to the paradox of output saturation. Therefore, in order to deal with constraints, BLF-based controllers should meet feasibility conditions [21], that is, the constraint variables should always be kept within the predefined boundaries. Parameters should also be designed appropriately because if the constraint is too small, it may be impractical. A lot of research has been conducted in [21,22] to eliminate strict feasibility conditions. In these works, the time derivative of the universal barrier function is regarded as an auxiliary system, and then incorporated into the control system [23].

On the basis of previous research, an adaptive output-feedback control method for manipulator systems with multiple constraints has been developed. A hybrid-state observer has been designed to estimate unmeasured states and disturbances in order to overcome the output constraint and input deadzone. An adaptive controller has been developed by combining back-stepping and BLF technology. It was proved that the proposed control method can ensure that all signals in the closed-loop system are bounded, simultaneously avoiding input and output constraints. The main contributions of the proposed control scheme are as follows: (i) the input deadzone and output constraints are solved using a BLF and auxiliary observer, respectively; (ii) the unmeasured state and disturbance are estimated by designing a hybrid state observer; and (iii) the designed system has a closed loop and stability.

## 2. Problem Description

The dynamic of a robot can be described as follows,
(1)$$M(q)\ddot{q}+C(q,\dot{q})\dot{q}+G(q)=\tau +\tau_{d}$$where $q\in R^{n}$ is the vector of joint displacements, $\tau \in R^{n}$ is the vector of joint torques supplied by the actuators, $M(q)\in R^{n}$ is the inertia matrix, $C(q,\dot{q})\in R^{n}$ is the Coriolis and centrifugal matrix, and $G(q)\in R^{n}$ is the gravitational force. $\tau_{d}$ is the uncertainty, including unmodeled dynamics and external disturbances. The inertia matrix M(q) is symmetric and positive definite, and the matrix $\dot{M}-2C(q,\dot{q})$ is skew-symmetric.

Considering parameter uncertainty, we define the accurate dynamic parameters are
(2)$$\begin{array}{l} M(q)=M(q)+\Delta M(q)\\ C(q,\dot{q})=C(q,\dot{q})+\Delta C(q,\dot{q})\\ G(q)=G(q)+\Delta G(q) \end{array}$$
where M(q), $C(q,\dot{q})$ and G(q) are the nominal parameter of the manipulator, and ΔM(q), $\Delta C(q,\dot{q})$, and ΔG(q) are the parameter uncertainty. The total unknown disturbance is defined as
(3)$$d=\tau_{d}-\Delta M(q)\ddot{q}-\Delta C(q,\dot{q})\dot{q}-\Delta G(q)$$

Usually, the disturbance *d* is bounded and continuous and satisfies $\left|{\dot{d}}_{i}\right|\leq d_{iM},i=1,2,3$, where $d_{iM}$ is the positive constant.

Then, the model of the manipulator can be rewritten as:(4)$$M(q)\ddot{q}+C(q,\dot{q})\dot{q}+G(q)=\tau +d$$

Letting $x_{1}=q$, $x_{2}=\dot{q}$, $x_{3}={\dot{x}}_{2}=\ddot{q}$, the model (4) may be rewritten as:(5)$$\left\{ \begin{array}{l} y=x_{1}\\ x_{2}={\dot{x}}_{1}\\ x_{3}=M^{-1}(q)\left[u(\tau )-G(q)-C(q,\dot{q})\dot{q}\right]+M^{-1}(q)d \end{array}\right.$$

Letting the desirable trajectory be $x_{d}(t)$, the actual output trajectory x(t) is
(6)$$\begin{array}{l} x_{d}(t)={\left[x_{d1}(t),x_{d2}(t),\cdot \cdot \cdot ,x_{dn}(t)\right]}^{T}\\ x(t)={\left[x_{1}(t),x_{2}(t),\cdots ,x_{n}(t)\right]}^{T} \end{array}$$

All the signals are bounded and we have $-k_{c1}\leq x(t)\leq k_{c1}$, where $k_{c1}=[k_{c11},k_{c11},\cdot \cdot \cdot ,k_{c11}{]}^{T}$ is a positive constant vector. In this model, the position of the manipulator can be measured, but the joint speed and acceleration cannot be measured directly. To achieve accurate control of this manipulator, it is necessary to estimate these unknown states.

For the manipulator, the actual inputs and outputs have various types of nonlinear constraints, such as output saturation, input deadzone, and hysteresis. $S=[S_{1}\cdots S_{n}]$ is the torque vector of the motor, and its output saturation may be represented as [24]:(7)$$S\left(\tau \right)=\left\{ \begin{array}{l} u\left(\tau_{i\max}\right)\;\mathrm{if}\;\tau_{i}>\tau_{i\max}\\ g_{rk}\left(\tau_{k}\right)\;\mathrm{if}\;0\leq \tau_{k}\leq \tau_{i\max}\\ g_{lk}\left(\tau_{i}\right)\;\mathrm{if}\;-\tau_{k\min}\leq \tau_{i}\leq 0\\ u\left(-\tau_{i\min}\right)\;\mathrm{if}\;\tau_{i}\leq -\tau_{i\min} \end{array}\right.$$
where $u\left(\tau_{i}\right)\max$ and $u\left(\tau_{i}\right)\min$ are the maximum and minimum of the input signals, respectively, and $g_{lk}$ and $g_{rk}$ are the nonlinear smooth continuous functions.

The input deadzone of the joint actuator may be represented as [25]:(8)$$U(\tau_{i})=\left\{ \begin{array}{l} D_{r}(\tau_{i}-b_{r}),\tau_{i}\geq b_{r}\\ 0,\;b_{l}<\tau_{i}<b_{r}\\ D_{l}(b_{l}-\tau_{i}),\tau_{i}\leq b_{l} \end{array}\right.,i=1,2,\cdots ,n$$
where $\tau_{i}$ is the input, $b_{l}$ and $b_{r}$ are the boundaries, and $D_{r}\left(\tau \right)$ and $D_{l}$ are the deadzone function.

The hysteresis nonlinearity may be represented as [26]:(9)$$\frac{d\phi_{i}(\tau_{i})}{dt}=h_{ai}|\frac{d\tau_{i}}{dt}|\left[h_{ai}\tau_{i}-\phi_{i}(\tau_{i})\right]+h_{bi}\frac{d\tau_{i}}{dt}$$$\phi (\tau )=H_{a}\tau_{i}+H_{b}d(\tau_{i}),i=1,2,\cdots ,n$

here
(10)$$\begin{array}{l} H_{a}=diag\{h_{a1},\cdots ,h_{an}\}>0,\\ H_{b}=diag\{h_{b1},\cdots ,h_{bn}\}>0,\\ d(\tau )={\left[d_{1}(\tau_{1}),d_{2}(\tau_{3}),\cdots ,d_{n}(\tau_{n})\right]}^{T}\leq d^{*} \end{array}$$

Obviously, this manipulator exists with unknown dynamics and an external disturbance. Moreover, some states cannot be measured directly, and many nonlinear constraints exist. Collecting all of this together, it is difficult to achieve accurate control of this manipulator.

## 3. Design of Controller

In order to achieve accurate control of the manipulator with multiple constraints, disturbances, and some unmeasurable states, a hybrid state/disturbance observer-based multiple-constraint control method is developed here. A hybrid state/disturbance observer is proposed to simultaneously estimate unmeasurable states and external disturbances. On this basis, a multiple-constraint control strategy is developed to achieve satisfactory control performance under multiple constraints. Moreover, the stability of the developed controller is analyzed and proved.

### 3.1. Hybrid State/Disturbance Observer

For the manipulator system (5), only the output state $x_{1}$ can be measured, while the other state $x_{2},x_{3}$ cannot be measured directly. To obtain the states and unknown disturbance of the manipulator system, a hybrid state/disturbance observer system is designed as follows:(11)$$\begin{array}{l} {\dot{\hat{x}}}_{1}=l_{1}{\left|{\hat{x}}_{1}-y\right|}^{\frac{2}{3}}\mathrm{sgn}({\hat{x}}_{1}-y)+{\hat{x}}_{2}-{\hat{d}}_{1}\\ {\dot{\hat{x}}}_{2}=l_{2}{\left|{\hat{x}}_{2}-{\dot{x}}_{1}\right|}^{\frac{1}{2}}\mathrm{sgn}({\hat{x}}_{2}-{\dot{x}}_{1})+{\hat{x}}_{3}-{\hat{d}}_{2}\\ {\dot{\hat{x}}}_{3}=l_{3}\mathrm{sgn}({\hat{x}}_{3}-{\dot{x}}_{2})-{\hat{d}}_{3} \end{array}$$
where ${\hat{x}}_{i}$ is the estimation of $x_{i}$ and ${\hat{d}}_{i}$ is the estimation of disturbance $d_{i}$.

(a)Solving the hybrid observer

In order to find the solution of this hybrid observer, a two-step solving algorithm is developed: (1) robustly estimate the states under the given disturbance, roughly estimated according to expert experience or experiment; and (2) estimate the disturbance under the given states.

Usually, the initial disturbance can be roughly estimated according to expert experience or experiment, set as ${\hat{d}}_{1}$.

Moreover, the disturbance is often bounded and has ${\tilde{d}}_{1}=\left|d_{1}-{\hat{d}}_{1}\right|\leq \xi$, in which ξ is the estimation error of the disturbance $d_{i}$.

When the parameters $l_{i}$ and the symmetric and positive definite matrices P and Q are properly chosen, the following inequality can be satisfied:(12)$$A^{T}P+PA\leq -Q,A=\left(\begin{array}{ccc} -l_{1} & & \\ \vdots & I_{i-1} & \\ -l_{i} & \cdots & 0 \end{array}\right),i=1,2,3$$

According to (3), $\left|{\dot{d}}_{i}\right|\leq d_{iM}$ and the following inequality will be held in finite time,
(13)$$\begin{array}{l} \left|{\hat{x}}_{i+1}-x_{i+1}\right|\\ =\left|l_{i}{\left|{\hat{x}}_{i}-{\dot{x}}_{i-1}\right|}^{\frac{3-i}{3-i+1}}\mathrm{sgn}({\hat{x}}_{i}-{\dot{x}}_{i-1})+{\hat{x}}_{i+1}-{\hat{d}}_{i}-x_{i+1}\right|\\ \leq \left|l_{i}{\left|{\hat{x}}_{i}-{\dot{x}}_{i-1}\right|}^{\frac{3-i}{3-i+1}}\mathrm{sgn}({\hat{x}}_{i}-{\dot{x}}_{i-1})-{\hat{d}}_{i}+\varepsilon \right|\\ \leq \left|l_{i}{\left|{\hat{x}}_{i}-{\dot{x}}_{i-1}\right|}^{\frac{3-i}{3-i+1}}\varepsilon +\varepsilon \right|+\left|{\hat{d}}_{i}\right|\\ \leq \left|l_{i}{\left|{\hat{x}}_{i}-{\dot{x}}_{i-1}\right|}^{\frac{3-i}{3-i+1}}\varepsilon +\varepsilon \right|+d_{iM},i=2,3 \end{array}$$

According to the inequality (13), in the presence of a bounded noise, the state error of the system is bounded.

In order to accurately estimate the disturbance, the following linear disturbance model [27] is used,
(14)$$\left\{ \begin{array}{l} \dot{W}=SW+Ed^{\left(r\right)}\\ d=LW \end{array}\right.$$
where $d^{\left(r\right)}$ denotes the $r^{th}$ derivatives of d and $S\in R^{2r\times 2r}$. The matrices S, E, and L have the following forms.
(15)$$S=\left[\begin{array}{cc} O_{(2r-2)\times 2} & I_{2r-2}\\ O_{2\times 2} & O_{2\times (2r-2)} \end{array}\right]E=\left[\begin{array}{c} O_{(2r-2)\times 2}\\ I_{2} \end{array}\right]L=\left[\begin{array}{cc} I_{2} & O_{2\times (2r-2)} \end{array}\right]$$

Combining (5) and (14), an extended system is obtained as,
(16)$$\left\{ \begin{array}{l} {\dot{x}}_{2}=f+LW\\ \dot{W}=SW+Ed^{\left(r\right)} \end{array}\right.$$
where $f=M_{0}{}^{-1}(q)\left[\tau -G_{0}(q)-C_{0}(q,\dot{q})\dot{q}\right]$. An auxiliary variable is introduced as follows:(17)$$O=\left[\begin{array}{c} O_{1}\\ O_{2} \end{array}\right]=\left[\begin{array}{c} x_{2}\\ p(x_{2})+W \end{array}\right]$$
where $p(s)=s^{r}+q_{1}s^{r-1}+\cdots +q_{r-1}s+q_{r}=0$ is a polynomial vector, which is designed as shown in [25].

By differentiating Equation (17), and with consideration of Equations (10), (15) and (17), one has
(18)$$\dot{O}=\left[\begin{array}{c} {\dot{O}}_{1}\\ {\dot{O}}_{2} \end{array}\right]=\left[\begin{array}{c} f+L(O_{2}+P(O_{1}))\\ (S-\frac{\partial P(O_{1})}{\partial O_{1}}L)O_{2}+(SP(O_{1})-\frac{\partial P(O_{1})}{\partial O_{1}}(f+LP(O_{1}))+Ed^{\left(r\right)} \end{array}\right]$$

From (18) and (14), the disturbance observer is designed as
(19)$$\left\{ \begin{array}{l} {\dot{\hat{O}}}_{2}=(S-\left(\partial P(O_{1}\right)/\partial O_{1})L)O_{2}+(SP(O_{1})-\left(\partial P(O_{1}\right)/\partial O_{1})(f+LP(O_{1}))\\ \hat{W}={\hat{O}}_{2}{+P(O}_{1})\\ \hat{d}=L\hat{W} \end{array}\right.$$
where $f=M^{-1}(q)\left[\tau -G(q)-C(q,\dot{q})\dot{q}\right]$.

According to (19), it is easy to obtain the disturbance. Obviously, when the initial estimation of the disturbance is accurate, using (12) and (13), it is easy to obtain the states, upon which it can effectively estimate the disturbance using (19). When the initial estimation of the disturbance is inaccurate, it uses an iterative process to obtain the states and disturbance:

**Step (1):** first, robustly estimate the states under the initial disturbance, which is roughly estimated according to expert experience or experiment;

**Step (2):** then, estimate the disturbance under the given states;

**Step (3):** use the estimated disturbance to replace the initial disturbance and repeat step (1), (2), and (3) until the satisfactory states and the disturbance is obtained.

(b)Performance analysis of the hybrid observer

The estimation errors are defined as
(20)$${\tilde{x}}_{1}={\hat{x}}_{1}-y,{\tilde{x}}_{2}={\hat{x}}_{2}-{\dot{x}}_{1},{\tilde{x}}_{3}={\hat{x}}_{3}-{\dot{x}}_{2},\tilde{d}=\hat{d}-d$$

According to Equation (20), the disturbance error $\tilde{d}$ may be represented as
(21)$$\tilde{d}=\hat{d}-d=L(W-\hat{W})$$

From (21), one has
(22)$$\begin{array}{l} \dot{\tilde{d}}=\hat{d}-d=L(\dot{W}-\dot{\hat{W}})=SW+Ed^{\left(r\right)}-(S-\frac{\partial P(O_{1})}{\partial O_{1}}L)O_{2}\\ -(SP(O_{1})-\frac{\partial P(O_{1})}{\partial O_{1}}(f+LP(O_{1}))-\dot{P}(O_{1})\\ \;=(S-\frac{\partial P(O_{1})}{\partial O_{1}}L)e+Ed^{\left(r\right)} \end{array}$$

According to (10), (11), (13), and (22), the error system of the hybrid observer is the following,
(23)$$\begin{array}{l} \dot{\tilde{x}}=A\tilde{x}+f+\tilde{d},\tilde{y}={\tilde{x}}_{1}\\ f=M^{-1}(q)\left[\tau -G(q)-C(q,\dot{q})\dot{q}\right] \end{array}$$
where $\tilde{x}={\left[\begin{array}{ccc} {\tilde{x}}_{1} & {\tilde{x}}_{2} & {\tilde{x}}_{3} \end{array}\right]}^{T}$, $\tilde{d}={\left[\begin{array}{ccc} {\tilde{d}}_{1} & {\tilde{d}}_{2} & {\tilde{d}}_{3} \end{array}\right]}^{T}=L(W-\hat{W})$.

From (11) and (23), one has
(24)$$\begin{array}{l} {\dot{\tilde{x}}}_{1}=l_{1}{\left|{\tilde{x}}_{1}\right|}^{\frac{2}{3}}sign({\tilde{x}}_{1})+{\tilde{x}}_{2}-{\hat{d}}_{1}\\ {\dot{\tilde{x}}}_{2}=l_{2}{\left|{\tilde{x}}_{2}\right|}^{\frac{1}{2}}sign({\tilde{x}}_{2})+{\tilde{x}}_{3}-{\hat{d}}_{2}\\ {\dot{\tilde{x}}}_{3}=l_{3}sign({\tilde{x}}_{3})-{\hat{d}}_{3}\\ \tilde{d}=Asign({\tilde{x}}_{i+1}-{\tilde{\dot{x}}}_{i})+M^{-1}(q)\left[\tau -G(q)-C(q,\dot{q})\dot{q}\right]-\hat{d}\; \end{array}$$

**Theorem 1.** 
*Using the designed hybrid observer (11) and the three-step solving algorithm, the state estimation error*

$\tilde{x}$

*and the disturbance estimation error*

$\tilde{d}$

*converge to the expected value in finite time, which ensures that*

$\tilde{x}$

*and*

$\tilde{d}$

*are bounded, guaranteeing*

$x_{i}(i=1,2,3)$

*will not escape to infinity before the finite time convergence of the hybrid observer error. As such, it can effectively estimate the states and disturbance.*


**Proof.** See Appendix A. □

### 3.2. Design of Feedback Controller

In order to achieve the well-tracking performance of the manipulator under multiple constraints, a multiple-constraint control strategy with consideration of the integrated barrier Lyapunov function method and the back-stepping algorithm is developed here, as shown in Figure 1. The tracking error is defined as $e={\left[e_{1},\cdots ,e_{n}\right]}^{T}=x_{1}-x_{d}=\left[x_{11}-x_{d1},\cdots ,x_{1n}-x_{dn}\right]$, where $x_{1}=\left[x_{11},x_{12},\cdot \cdot \cdot ,x_{1n}\right]$ is the actual trajectory and $x_{d}$ is the desired trajectory. The back-stepping algorithm defines
(25)$$z={\left[z_{1},\cdots ,z_{n}\right]}^{T}=x_{2}-\alpha$$
where α denotes the virtual control variable.

The derivative of (25) is
(26)$$\dot{z}={\dot{x}}_{2}-\dot{\alpha }$$

$x_{0}$ is a small positive constant, which satisfies
(27)$$\left\{ \begin{array}{l} \left|x_{1i}\right|\leq k_{c1i}\leq k_{ai},i=1,2,\cdot \cdot \cdot ,n\\ \left|x_{di}\right|\leq k_{c1i}\leq k_{ai},i=1,2,\cdot \cdot \cdot ,n \end{array}\right.$$
where $k_{ai}=k_{c1i}+x_{0}$.

The following two-step strategy is developed to design the controller.

**Step 1:** For the multi-link manipulator, construct the following Lyapunov function,
(28)$$V_{1}=\sum_{i=1}^{n}\int_{0}^{e_{i}}\frac{\sigma k_{ai}^{2}}{k_{ai}^{2}-{\left(\sigma +x_{di}\right)}^{2}}d\sigma$$
where σ is a modification value.

Differentiating (28) yields
(29)$${\dot{V}}_{1}=\sum_{i=1}^{n}\frac{k_{ai}^{2}({\hat{z}}_{i}+\alpha_{i})}{k_{ai}^{2}-x_{1i}^{2}}-\sum_{i=1}^{n}e_{i}\rho_{i}{\dot{x}}_{di}$$
here,
(30)$$\begin{array}{l} \rho_{i}(e_{1i},x_{di})=\int_{0}^{1}\frac{k_{ai}^{2}}{k_{ai}^{2}-{\left(\beta e_{i}+x_{di}\right)}^{2}}d\beta \\ =\frac{k_{ai}}{2e_{1i}}\ln\frac{\left(k_{ai}+e_{i}+x_{di})(k_{ai}-x_{di}\right)}{\left(k_{ai}-e_{i}-x_{di})(k_{ai}+x_{di}\right)} \end{array}$$

$\rho_{i}(e_{1i},x_{di})$ is well defined and bounded around the neighborhood of $e_{1i}=0$.

Using L’Hopital’s rule, one has
(31)$$\begin{array}{l} \underset{e_{1i}\to 0}{\lim}\rho_{i}(e_{1i},x_{di})=\underset{e_{1i}\to 0}{\lim}\frac{k_{ai}}{2e_{1i}}\ln\frac{\left(k_{ai}+e_{i}+x_{di})(k_{ai}-x_{di}\right)}{\left(k_{ai}-e_{i}-x_{di})(k_{ai}+x_{di}\right)}\\ =\frac{k_{ai}^{2}}{k_{ai}^{2}-x_{di}^{2}} \end{array}$$

Design virtual control variable $\alpha_{i}$ as follows,
(32)$$\alpha_{i}=(-k_{1i}e_{1i}+\frac{(k_{ai}^{2}-x_{1i}^{2}){\dot{x}}_{di}\rho_{i}}{k_{ai}^{2}}),i=1,2,\cdots ,n$$
where $k_{ai}$ is the positive control gain.

Combining (29) with (30), one has
(33)$${\dot{V}}_{1}=-\sum_{i=1}^{n}\frac{k_{1i}k_{ai}^{2}e_{1i}^{2}}{k_{ai}^{2}-x_{1i}^{2}}+\sum_{i=1}^{n}\frac{k_{ai}^{2}e_{1i}{\hat{z}}_{i}}{k_{ai}^{2}-x_{1i}^{2}}$$

**Step 2:** Establish the second Lyapunov function as follows
(34)$$V_{2}=V_{1}+\frac{1}{2}z_{}^{T}M(x_{1})z$$

Differentiating $V_{2}$ yields,
(35)$${\dot{V}}_{2}={\dot{V}}_{1}+z_{}^{T}M(x_{1})\dot{z}+z_{}^{T}\frac{1}{2}\dot{M}(x_{1})\dot{z}$$

Substituting (5) and (25) into (35), we have
(36)$$\begin{array}{l} {\dot{V}}_{2}={\dot{V}}_{1}+z_{}^{T}\left[M(x_{1})({\dot{x}}_{2}-\dot{\alpha })+\frac{1}{2}\dot{M}(x_{1})\dot{\hat{z}}\right]\\ ={\dot{V}}_{1}+z_{}^{T}[\tau -d-C(x_{1},x_{2})\alpha +G(x_{1})-M(x_{1})\dot{\alpha })\\ \;+\frac{1}{2}(\dot{M}(x_{1})-2C(x_{1},x_{2}))e_{2}] \end{array}$$

Substituting (32) and (34) into (35) and (36) is simplified as
(37)$$\begin{array}{l} {\dot{V}}_{2}=-\sum_{i=1}^{n}\frac{k_{1i}k_{ai}^{2}e_{i}^{2}}{k_{ai}^{2}-x_{1i}^{2}}+\sum_{i=1}^{n}\frac{k_{ai}^{2}e_{i}z_{i}}{k_{ai}^{2}-x_{1i}^{2}}\\ +z^{T}\left[\tau -G(x_{1})+C(x_{1},x_{2})\alpha +M(x_{1})\dot{\alpha }-\hat{d}\right] \end{array}$$

In order to make ${\dot{V}}_{2}<0$, considering the hysteresis (9) and input saturation (7), the torque control law is designed as follows,
(38)$$\tau =-\sum_{i=1}^{n}\frac{k_{ai}^{2}e_{i}z_{i}}{k_{ai}^{2}-x_{1i}^{2}}-Kz+G(x_{1})+C(x_{1},x_{2})\alpha +M(x_{1})\dot{\alpha }+\hat{d}$$
where $K=diag(k_{1},k_{2})$ is a positive gain matrix, and it must satisfy
(39)$$\min\left\{2\lambda_{\min}(K)/\lambda_{\max}(M(x_{1}))\right\}>0$$
where $\lambda_{\min}(•)$ and $\lambda_{\max}(•)$ denotes the maximum and minimum eigenvalue of (•).

Then, considering the input deadzone, and by combining the input constraint (8) and the control law (39), the final control law is designed as follows,
(40)$$\begin{array}{l} u(\tau )=-Kz-\sum_{i=1}^{n}\frac{k_{ai}^{2}e_{i}z_{i}}{k_{ai}^{2}-x_{1i}^{2}}\\ \;+H_{a}^{}\left[G_{0}(x_{1})+C_{0}(x_{1},x_{2})\alpha_{1}+M_{0}(x_{1}){\dot{\alpha }}_{1}\right]+H_{b}^{}\hat{d} \end{array}$$

Therefore, the control law can deal with all constraints, including hysteresis, output saturation, and input deadzone.

**Theorem 2.** 
*Considering the manipulator system (5) with an unknown disturbance, input deadzone, and output saturation, and given feedback control laws (38) and (40), the closed-loop system is semi-globally stable.*


**Proof.** Substituting (38) into (37) yields
(41)$${\dot{V}}_{2}=-\sum_{i=1}^{n}\frac{k_{1i}k_{ai}^{2}e_{i}^{2}}{k_{ai}^{2}-x_{1i}^{2}}-z^{T}K\hat{z}$$Define $\hat{z}=\tilde{z}+z$ and substitute it into (41)
(42)$${\dot{V}}_{2}=-\sum_{i=1}^{n}\frac{k_{1i}k_{ai}^{2}e_{i}^{2}}{k_{ai}^{2}-x_{1i}^{2}}-z^{T}Kz-z^{T}K\tilde{z}$$While
(43)$$\begin{array}{l} -z^{T}K\tilde{z}\leq \frac{1}{2}z^{T}\tilde{z}+\frac{1}{2}{\left(K\tilde{z}\right)}^{T}\left(K\tilde{z}\right)\\ -z^{T}\tilde{z}\leq \frac{1}{2}z^{T}\tilde{z}+\frac{1}{2}{\tilde{z}}^{T}\tilde{z} \end{array}$$From (42) and (43), one has
(44)$$\begin{array}{l} {\dot{V}}_{2}=-\sum_{i=1}^{n}\frac{k_{1i}k_{ai}^{2}e_{i}^{2}}{k_{ai}^{2}-x_{1i}^{2}}-z^{T}Kz-z^{T}\tilde{z}-z^{T}K\tilde{z}\\ \;\leq -\sum_{i=1}^{n}\frac{k_{1i}k_{ai}^{2}e_{i}^{2}}{k_{ai}^{2}-x_{1i}^{2}}+\frac{1}{2}z^{T}\tilde{z}+\frac{1}{2}{\left(K\tilde{z}\right)}^{T}\left(K\tilde{z}\right)-\frac{1}{2}z^{T}\tilde{z}+\frac{1}{2}{\tilde{z}}^{T}\tilde{z}\\ \;=-\sum_{i=1}^{n}\frac{k_{1i}k_{ai}^{2}e_{i}^{2}}{k_{ai}^{2}-x_{1i}^{2}}-z^{T}Kz\leq -V_{2}\rho \end{array}$$Because
(45)$$\rho =\min\left\{\min_{a\leq i\leq n}(k_{i})\right\}=\min\left\{2\lambda_{\min}(k_{2})/\lambda_{\max}(M)\right\}>0.$$
one has ${\dot{V}}_{2}<0$ according to (44) and (45). This indicates that the system tends to be stable in a small spectrum. □

## 4. Case Studies and Experiment

In this section, we simulated the proposed algorithm and validated the proposed control algorithm on a robot platform consisting of a six-degree-of-freedom (DOF) robot (JAKA ZU-7s), DC power, and an embedded DSP control system, as depicted in Figure 1. We carried out the simulation and experiment on a 2-DOF and 6-DOF robot platform, respectively. The dynamic model of the 2-DOF robot and its parameters are shown in Equation (46) and Table 1. The parameters of the 6-DOF robot are shown in Table 2. The accurate dynamic model of the 6-DOF robot was obtained by parameter identification.

For the 2-DOF manipulator, the inertia matrix M(q), the Coriolis and centrifugal matrix $C(q,\dot{q})$, and the gravity matrix G(q) are as follows
(46)$$\begin{array}{l} M(q)=\left[\begin{array}{l} M_{11}M_{12}\\ M_{21}M_{22} \end{array}\right],M_{11}=m_{1}l_{1}^{2}+m_{1}(l_{1}^{2}+l_{2}^{2}+2l_{1}l_{2}{\cos(q}_{2}))\\ M_{12}{=m}_{2}(l_{2}^{2}+l_{1}l_{2}{\cos(q}_{2})),M_{21}{=m}_{2}(l_{2}^{2}+l_{1}l_{2}{\cos(q}_{2}))\;,M_{22}{=m}_{2}l_{2}^{2}\\ C(q,\dot{q})=\left[\begin{array}{l} {-m}_{2}l_{1}l_{2}^{}{\dot{q}}_{2}{\sin(q}_{2}{)\;-m}_{2}l_{1}l_{2}^{}{(\dot{q}}_{1}+{\dot{q}}_{2}{)\sin(q}_{2})\\ {-m}_{2}l_{1}l_{2}^{}{\dot{q}}_{1}{\sin(q}_{2})\;0 \end{array}\right]\\ G(q)=\left[\begin{array}{l} \left(m_{1}l_{2}+m_{2}l_{1})g\cos(q_{1})+m_{2}l_{2}{\mathrm{gcos}(q}_{1}{+q}_{2}\right)\\ m_{2}l_{2}{\mathrm{gcos}(q}_{1}{+q}_{2}) \end{array}\right] \end{array}$$

The initial position is $q_{0}=\left[0\;,\;0\right]$ and ${\dot{q}}_{0}=\left[0\;,\;0\right]$. The unknown dynamics and external disturbance are defined as follows:(1)Unknown unmodeled dynamics
(47)$$\begin{array}{l} \Delta M(q)=0.2M(q)\\ \Delta C(q,\dot{q})=0.2C(q,\dot{q})\\ \Delta G(q)=0.2G(q) \end{array}$$

(2)Unknown external disturbance term


(48)
$$\tau_{d}=\left[\begin{array}{l} 10(1-\exp(-0.28\cdot t))(\sin(0.5\cdot \mathrm{pi}\cdot t))\;\\ 10(1-\exp(-0.28\cdot t))(\cos(0.5\cdot \mathrm{pi}\cdot t)) \end{array}\right]$$


Thus, the total disturbance is shown below,
(49)$$d=\tau_{d}-\Delta M(q)\ddot{q}-\Delta C(q,\dot{q})\dot{q}-\Delta G(q)$$

The input dead time and the output saturation are described as follows: $b_{r}=2.5$, $b_{l}=-4.5$, respectively, with
(50)$$U(\tau_{i})=\{\begin{array}{c} D_{r}(\tau )=2(\tau -b_{r})(\sin(\tau )+1)\\ D_{l}(\tau )={\left(\tau -b_{l}\right)}^{3} \end{array}$$

### 4.1. Control Performance

The desired trajectory and its velocity and acceleration are given as follows
(51)$$\begin{array}{l} \left[\begin{array}{c} q3\\ q4 \end{array}\right]=\left[\begin{array}{l} \sin(0.5t)+2\cdot \cos(t)\\ 2\cdot \cos(0.5t) \end{array}\right]\\ \left[\begin{array}{c} dq3\\ dq4 \end{array}\right]=\left[\begin{array}{l} 0.5\cos(0.5t)-2\sin(t)\\ -\sin(0.5t) \end{array}\right]\\ \left[\begin{array}{c} ddq3\\ ddq4 \end{array}\right]=\left[\begin{array}{l} -0.25\sin(0.5t)-2\cdot \cos(t)\\ -\sin(0.5t) \end{array}\right] \end{array}$$

Then, the proposed control method is used to control this manipulator with input nonlinearity, output saturation, and external disturbance states to track the reference trajectory.

The hybrid state/disturbance observer is the following,
(52)$$\begin{array}{l} {\dot{\hat{x}}}_{1}=-l_{1}{\left|{\hat{x}}_{1}-y\right|}^{\frac{2}{3}}\mathrm{sgn}({\hat{x}}_{1}-y)+{\hat{x}}_{2}\cdot v_{1}+d_{1}\\ {\dot{\hat{x}}}_{2}=-l_{3}{\left|{\hat{x}}_{1}-y\right|}^{\frac{1}{2}}\mathrm{sgn}({\hat{x}}_{2}-{\dot{x}}_{1})+d_{2}\\ {\dot{\hat{x}}}_{3}=-l_{3}\mathrm{sgn}({\hat{x}}_{3}-{\dot{x}}_{2})+d_{3} \end{array}$$
where $l_{1}=10$, $l_{2}=15$, and $l_{3}=15$. The initial disturbance is set as $d_{1}=0$, and the parameters of the disturbance observer are decided as follows
(53)$$\begin{array}{l} \begin{array}{ccc} S=\left[\begin{array}{cc} O_{2} & I_{2}\\ O_{2} & O_{2} \end{array}\right] & E=\left[\begin{array}{c} O_{2}\\ I_{2} \end{array}\right] & L=\left[\begin{array}{cc} I_{2} & O_{2} \end{array}\right] \end{array}\\ \begin{array}{ccc} l=\left[\begin{array}{c} 40\\ 0\\ 400\\ 0 \end{array}\;\begin{array}{c} 0\\ 40\\ 0\\ 400 \end{array}\right] & p=\left[\begin{array}{c} 40x_{2}\\ 400x_{2} \end{array}\;\right] & \end{array} \end{array}$$

The control gains are decided as $k_{a1}=k_{a2}=100$ and $K=diag[k_{1},k_{2}]=diag\left[5,5\right]$.

Using the designed observer, the estimated state $x_{2}$, $x_{3}$ and its observation error are shown in Figure 2a–d, respectively. Figure 3 show the disturbance estimation and its estimation error, respectively. From these figures, the unmeasurable state value of each joint and the disturbance can be accurately estimated.

Using this designed controller, the control torque is shown in Figure 4, the desired trajectory and the practical trajectory are shown in Figure 5, and their difference is shown in Figure 6. From these figures, it can be seen that the control torque is smooth without a sudden change, it does not violate output constraints, and the proposed method can effectively achieve the tracking of this manipulator, even if there are input nonlinearity, output saturation, external disturbances, and unmeasurable states.

Further, a comparison between the proposed method and three commonly used control methods, i.e., the neural network adaptive control [25], robust adaptive control [26], and fuzzy adaptive control [27], is carried out. The trajectory track and tracking error under the different control methods are shown in Figure 7, Figure 8, and Table 3, respectively. From these figures, it can be seen that the proposed control method has a smaller relative tracking error and deviation angle than the other ones. This is because the proposed method considers input nonlinearity, output saturation, and external disturbances together, while the other methods do not. Although these common methods have excellent performance without constraints, they cannot deal with these constraints when there are input deadzones and output saturation constraints. Compared with the methods proposed in this paper, the trajectory-tracking control accuracy is relatively lower.

### 4.2. Experiment

To further validate the proposed method, an experiment based on the JAKA robot platform experiment was conducted. The layout of the robot manipulator is shown in Figure 1. The type of manipulator is a collaborative ultralight robot, JAKA zu7s. The position model controls the joints in Cartesian space, while the velocity model and torque model control the joints in angular space. The signals are sent to the DSP controller through a USB port by MATLAB R2018b and by compiling the program. The execution frequency and sampling frequency of the controller are 100 Hz and 20 Hz, respectively. The following is the reference input trajectory
(54)$$\left[\begin{array}{c} q1\\ q2\\ q3\\ q4\\ q5\\ q6 \end{array}\right]=\left[\begin{array}{c} 4\sin(\frac{\pi }{2}t)+3\cos(\pi t)\\ 3\cos(\frac{\pi }{2}t)+0.2\sin(t)\\ 4\sin(\frac{\pi }{2}t)+2\cos(\pi t)\\ 3\cos(\frac{\pi }{2}t)\\ \sin(\frac{\pi }{5}t)+2\cos(\frac{2}{5}\pi t)\\ 2\cos(\frac{\pi }{5}t) \end{array}\right]$$

The parameters of the deadzone are defined as b_r_ = 5, b_l_ = −5, h_r_ = 15, and h_l_ = −10. The experimental results are shown in Figure 9, which shows that all joints have good tracking performance, and the proposed constraints are guaranteed by the experimental setup. For all joints, the average tracking error is within 0.01 degree, and the trajectories converge within the constraint range. Thus, the proposed method can realize highly precious tracking control.

## 5. Conclusions

A hybrid state/disturbance observer-based multiple-constraint control method for a manipulator is developed and evaluated in this study. The proposed hybrid state/disturbance observer can accurately estimate unmeasurable states and external disturbances. Combining back-stepping and BLF approaches, we propose an adaptive controller that can achieve tracking performance in the presence of several constraints and unknown disturbances. Simulations on a 2-DOF robot proved that the proposed strategy is effective. In comparison to several commonly employed intelligent control approaches, its control performance is superior. Experiments on a 6-DOF robot showed that all joints have good tracking performance and do not violate constraints. Future research will focus on the tracking control of underactuated systems with deferred constraints.

## Figures and Tables

**Figure 1 sensors-22-09112-f001:**
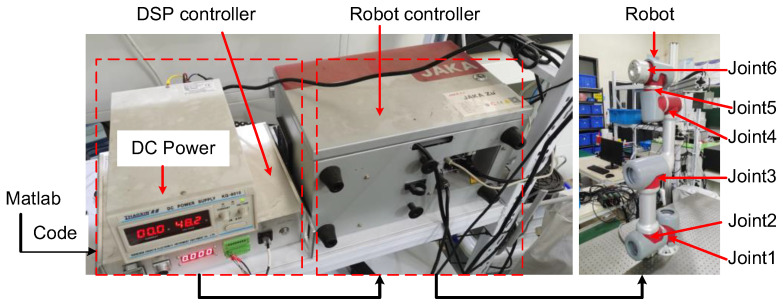
Robot experimental platform.

**Figure 2 sensors-22-09112-f002:**
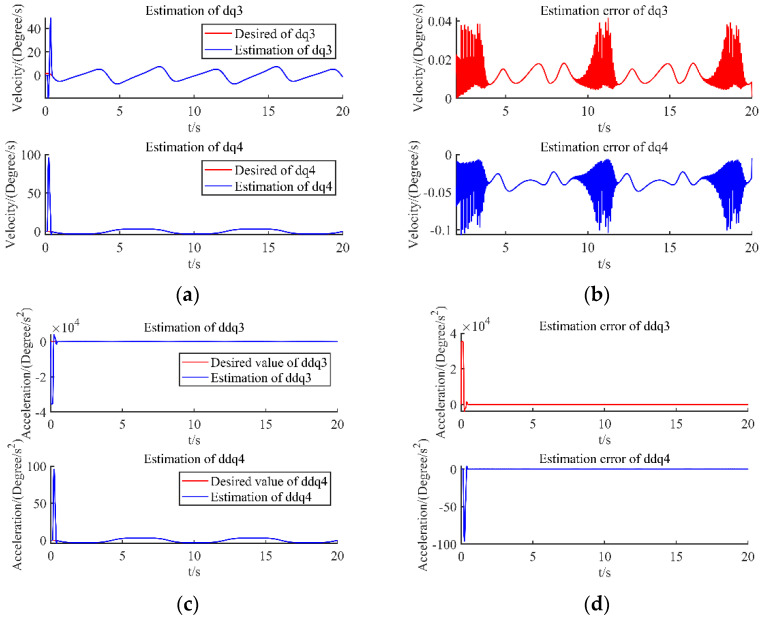
(**a**) State $x_{2}$ estimation; (**b**) State $x_{2}$ estimation errors; (**c**) State $x_{3}$ estimation; (**d**) State $x_{3}$ estimation errors.

**Figure 3 sensors-22-09112-f003:**
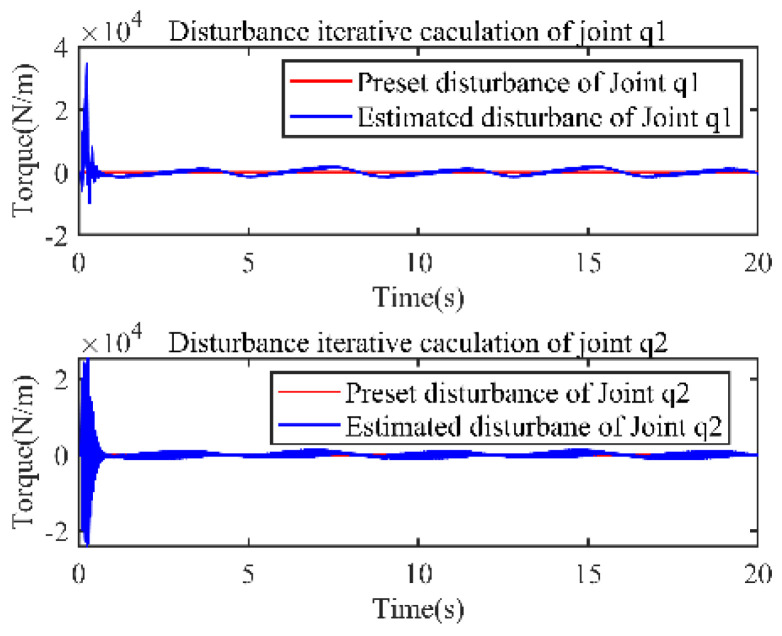
Disturbance observation.

**Figure 4 sensors-22-09112-f004:**
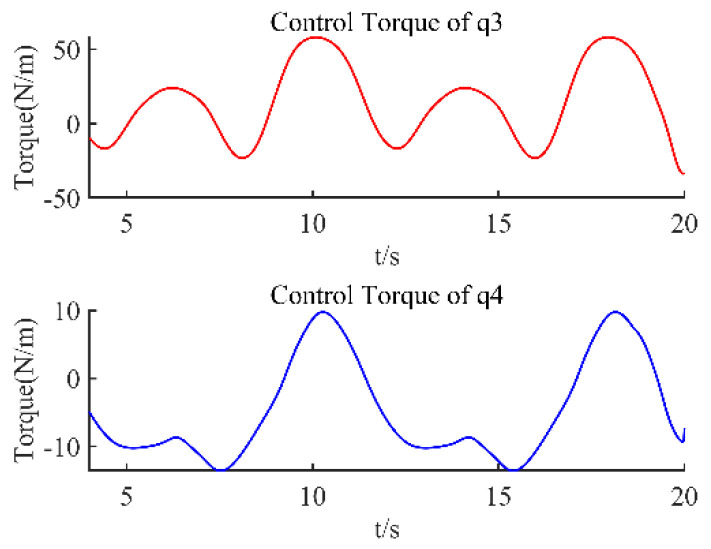
Joint’s control torque.

**Figure 5 sensors-22-09112-f005:**
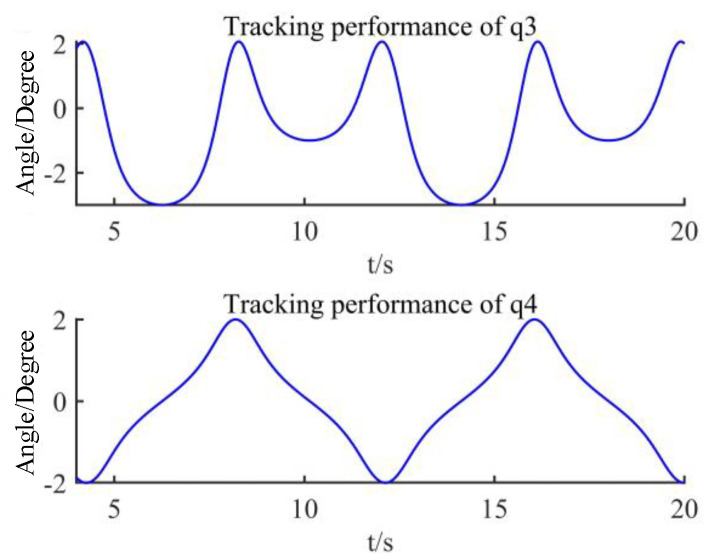
Joint’s tracking performance.

**Figure 6 sensors-22-09112-f006:**
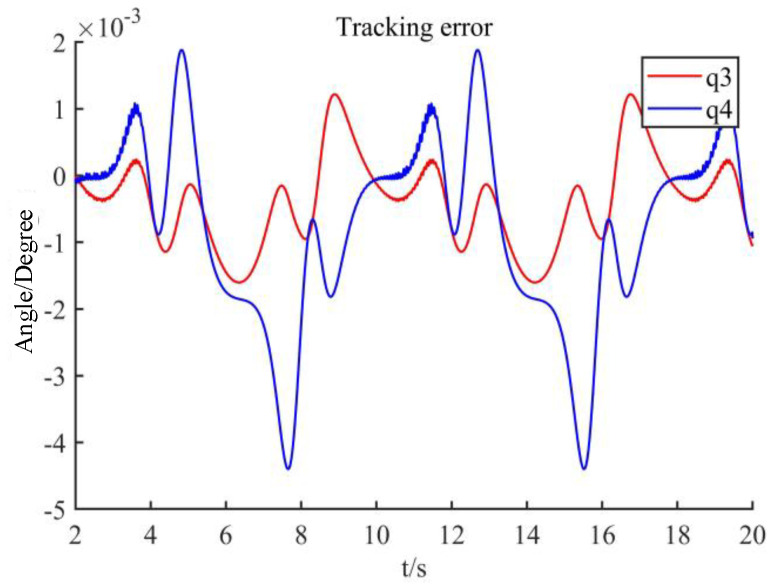
Tracking error.

**Figure 7 sensors-22-09112-f007:**
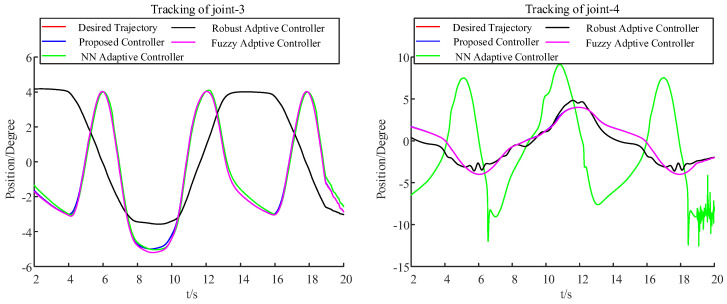
Trajectory tracking under different methods.

**Figure 8 sensors-22-09112-f008:**
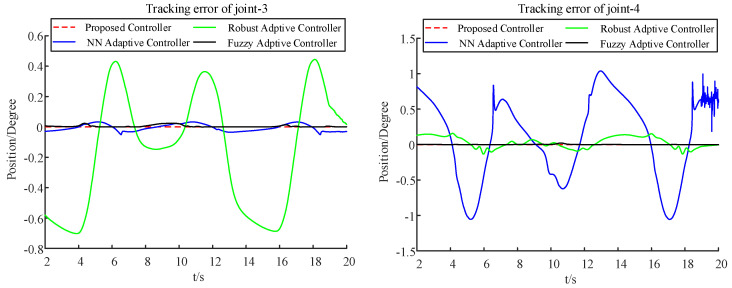
Tracking error under different methods.

**Figure 9 sensors-22-09112-f009:**
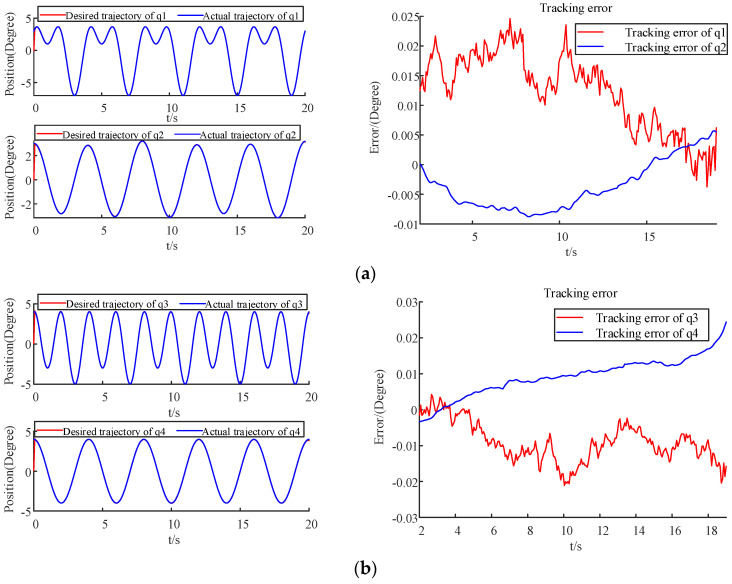
Tracking performance and errors for each joint of the 6DOF robot. (**a**) Tracking performance and tracking error of *q*1 and *q*2; (**b**) Tracking performance and tracking error of *q*3 and *q*4; (**c**) Tracking performance and tracking error of *q*5 and *q*6.

**Table 1 sensors-22-09112-t001:** Parameters of 2-DOF robot.

Parameters	Value	Parameters	Value
$m_{1}$	1.950 kg	$l_{1}$	303.5 mm
$m_{2}$	1.835 kg	$l_{2}$	208 mm

**Table 2 sensors-22-09112-t002:** D-H Parameters of 6-DOF JAKA robot.

Link *i*	$a_{i-1}\;(\mathrm{mm})$	$\alpha_{i-1}\;({}^{\circ})$	$d_{i}\;(\mathrm{mm})$	$q_{i}\;$
1	0	0	143.4	$q_{1}$
2	0	90.0	0	$q_{2}$
3	360.0	0	0	$q_{3}$
4	303.5	0	−115.0	$q_{4}$
5	0	90.0	113.5	$q_{5}$
6	0	−90.0	107.0	$q_{6}$

**Table 3 sensors-22-09112-t003:** Performance comparison.

Methods	Mean Absolute Error (Degree)	
	Joint 3 (*q*3)	Joint 4 (*q*4)
Proposed Control	0.0114	0.0087
NN Adaptive Control	0.2144	2.1308
Robust Adaptive Control	1.5823	0.3331
Fuzzy Adaptive Control	0.1680	0.0281

## Data Availability

Not applicable.

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
