# Peer review of "A Hybrid State/Disturbance Observer-Based Feedback Control of Robot with Multiple Constraints"

_sensors, 2022, doi:10.3390/s22239112_

Round 1

Reviewer 1 Report

How to realize a robust observer for tracking control, it is an interesting research topic. The paper presented authors’ research results in the area, which should be helpful and instructive for the researchers in the tracking control.

However, there are some issues with the paper that should not be overlooked, including.

1.      In the section about experiment in Chapter-4, there are no parameters about the test system, such as the moment of inertia and electrical parameters of the motors. The specifications of the system tested should be added to facilitate readers’ analysis and comparison;

2.      The author uses the trajectory represented by Equation 52 for the experiment. This trajectory is not complicated, why choose this trajectory for testing?

3.      What is the significance of Fig 4? Want to reflect the trajectory capabilities of the control system?

4.      In Fig 5, it seems that joint-3 and joint-4 are called q1 and q2 in some places, and dq1 and dq2 in some places. The author should explain all symbols used and unify the usage of symbols.

5.      In the 2nd figure of Fig 5, is the header "Desired trajectory qd1 and Actual trajectory of q2" a misspelling of "Desired trajectory dq2 and actual trajectory of q2"? A similar situation occurs in the 1st figure

Author Response

Response to Reviewer 1 Comments

Point 1

How to realize a robust observer for tracking control, it is an interesting research topic. The paper presented authors’ research results in the area, which should be helpful and instructive for the researchers in the tracking control. However, there are some issues with the paper that should not be overlooked, including.

Response 1:

Thanks for your positive comment.

Point 2

In the section about experiment in Chapter-4, there are no parameters about the test system, such as the moment of inertia and electrical parameters of the motors. The specifications of the system tested should be added to facilitate readers’ analysis and comparison;

Response 2:

Thanks for your helpful comments. The specifications of the system are added.

Point 3

The author uses the trajectory represented by Equation 52 for the experiment. This trajectory is not complicated, why choose this trajectory for testing?

Response 3:

Thanks for your helpful comments. The provided trajectory is the type of trajectory often encountered in actual working conditions, which can realize small range movement under multiple constraints.

Point 4

What is the significance of Fig 4? Want to reflect the trajectory capabilities of the control system?

Response 4:

Thanks for your comments. Yes, the significance of Fig 4 is that show the control torque of the controller to realize the trajectory tracking ability under constraints.

Point 5

In Fig 5, it seems that joint-3 and joint-4 are called q1 and q2 in some places, and dq1 and dq2 in some places. The author should explain all symbols used and unify the usage of symbols.

Response 5:

Thanks for your comments. Joint-3 and joint-4 are defined as q3 and q4 in paper, we have changed the symbol definition.

Point 6

In the 2nd figure of Fig 5, is the header "Desired trajectory qd1 and Actual trajectory of q2" a misspelling of "Desired trajectory dq2 and actual trajectory of q2"? A similar situation occurs in the 1st figure

Response 6:

Thanks for your positive comments. In order to avoid misunderstanding, we have defined the symbols in the text and figures again.

Again, the authors really appreciate the helpful suggestions and comments from Editor and reviewers.

Reviewer 2 Report

This paper develops a multiple-constraints control strategy to achieve the satisfactory control performance under the multiple constraints, such as hysteresis, input dead-zone and output saturation. Moreover, the stability of the developed controller is analyzed and proved. This paper is well written and organized, and contains some interesting results. However, the following comments need to be considered:

1、Compared with the existing results, the innovative display of this paper is not prominent, and further improvement is recommended.

2、It is recommended to add future research directions.

3、It is necessary to point out the shortcomings of the methods proposed in this paper.

4、There are many grammar and spelling mistakes that need to be further corrected.

Author Response

Response to Reviewer 2 Comments

Point 1:

This paper develops a multiple-constraints control strategy to achieve the satisfactory control performance under the multiple constraints, such as hysteresis, input dead-zone and output saturation. Moreover, the stability of the developed controller is analyzed and proved. This paper is well written and organized, and contains some interesting results. However, the following comments need to be considered:

Response 1:

Thanks for your positive comment.

Point 2:

Compared with the existing results, the innovative display of this paper is not prominent, and further improvement is recommended.

Response 2:

Thanks for your helpful comments. Thanks for your valuable comment. This comment lets us sense that our original paper still had shortcomings and did not demonstrate the value of the proposed method effectively. This gives us a great motivation to improve the quality of this paper and highlight the novelties and contributions.

The main contribution of this paper as follows: 1. A hybrid state/disturbance observer to estimate simultaneously the unmeasurable states and external disturbance. 2. an integral barrier Lyapunov function (IBLF) is proposed and implemented to handle output saturation constraints, and a back-stepping control strategy is designed to provide sufficient control performance under multiple constraints. According to these descriptions, we deeply revised and reorganized the Introduction to highlight the contributions that the paper brings to the state-of-the-art subject.

Point 3:

It is recommended to add future research directions.

Response 3:

Thanks for your comments.

In the future, we will investigate the tracking control of underactuated systems subject to deferred constraints, the future research directions have been added in paper.

Point 4:

It is necessary to point out the shortcomings of the methods proposed in this paper.

Response 4:

Thanks for your positive comments.

Since the method proposed in this paper considers many issues, it causes computational complexity when controlling a 6-DOF robot, leading to real-time problems in the control of the robot.

Point 5:

There are many grammar and spelling mistakes that need to be further corrected.

Response 5:

This paper has been careful checked and deeply revised, including language and typography.

Again, the authors really appreciate the helpful suggestions and comments from Editor and reviewers.

Reviewer 3 Report

The investigation is consisted of regular controller structure such as adaptive control of neural networks and fuzzy controllers. Part of 6 DOF robot part 2 DOF robotic system is simple to apply on these general structure.  The proposed control method should be applied for trajectory control of 6 DOF for testing performance.  basic dynamic and kinematic parameters of the robotic system should be given with table.

The investigation is consisted of regular controller structure such as adaptive control of neural networks and fuzzy controllers. Part of 6 DOF robot part 2 DOF robotic system is simple to apply on these general structure.  The proposed control method should be applied for trajectory control of 6 DOF for testing performance.  basic dynamic and kinematic parameters of the robotic system should be given with table.

The authors have been tried to design a proposed control for tracking control of n-DOF robot manipulator.
But they only applied to 2 DOF robot manipülatör.  This robot manipultor is very simple. The disturbances is very weak for testing the performance of the proposed tracking control system.

The propsed tracking is a proposed original control , but it should be tested on multi degrees of robot manipulators.
The compared controllers have good performance to control tracking of the DOF robot manipultor. Such as adaptive neuarl network controller.

They tried to compare with standart controllers such Adaptive Fuzzy Controllers, Adaptive Neural Controllers.But these controllers have suprieor performance to control tracking of 2 DOF robot manipulators.

The method should be implement experimentaly to multi degrees of robot manipulators such as 6 DOF. What further controls should be considered?

Dynamic and kinematic parameters of the robot manipultor should be given in Table.

Author Response

Response to Reviewer 3 Comments

Point 1:

The investigation is consisted of regular controller structure such as adaptive control of neural networks and fuzzy controllers. Part of 6 DOF robot part 2 DOF robotic system is simple to apply on these general structure.  The proposed control method should be applied for trajectory control of 6 DOF for testing performance.  basic dynamic and kinematic parameters of the robotic system should be given with table.

Response 1:

Thanks for your positive comment.

Point 2:

The authors have been tried to design a proposed control for tracking control of n-DOF robot manipulator. But they only applied to 2 DOF robot manipulator. This robot manipulator is very simple. The disturbances are very weak for testing the performance of the proposed tracking control system.

Response 2:

Thanks for your positive comments. Your view is right. Though 2-DOF robot manipulator is simple, this paper studies the influence of constraints on robot control, which is applicable to n-DOF robot. We have applied the proposed method on the 6-DOF robot.

Point 3:
The proposed tracking is a proposed original control, but it should be tested on multi degrees of robot manipulators.

Response 3:

Thanks for your comments. Your consideration is helpful, due to the complex calculation of the designed controller, it is difficult to achieve real-time control for the multi degree of freedom robot, resulting in a large delay. Therefore, in order to reduce the computational complexity and ensure the real-time control of the robot, this paper uses a two degree of freedom robot for simulation. The control experiment on 6-DOF robot is carried out, the corresponding results have been added to the paper.

Point 4:
The compared controllers have good performance to control tracking of the DOF robot manipulator. Such as adaptive neural network controller.

Response 4:

Thanks for your comments. Because neural network has the ability of high-precision unknown nonlinear approximation, it can approximate the unknown state variables to ensure the good control performance of the robot. Compared with the method proposed in this paper, the tracking performance is improved due to the high-precision observation of unknown state variables and disturbances, and the processing of input dead zones and input saturation. Therefore, the method proposed in this paper has better control performance.

Point 5:
They tried to compare with standard controllers such Adaptive Fuzzy Controllers, Adaptive Neural Controllers. But these controllers have superior performance to control tracking of 2 DOF robot manipulators.

Response 5:

Thanks for your positive comments. Although these methods have excellent performance without constraints, they cannot deal with these constraints when there are input dead zones and output saturation constraints. Compared with the methods proposed in this paper, the trajectory tracking control accuracy is relatively lower.

Point 6:
The method should be implemented experimentally to multi degrees of robot manipulators such as 6 DOF. What further controls should be considered?

Response 6:

Thanks for your positive comments. The proposed method has been implemented experimentally to the 6-DOF robot. Underactuated systems are very common in practice and it is significant to address their control problems under some complex nonlinearities such as modeling uncertainty and output and velocity constraints, etc. In the future, we will investigate the tracking control of underactuated systems subject to deferred constraints.

Point 7:
Dynamic and kinematic parameters of the robot manipulator should be given in Table.

Response 7:

Thanks for your valuable comments. Dynamic and kinematic parameters of the robot have been added in Section 4.

Again, the authors really appreciate the helpful suggestions and comments from Editor and reviewers.

Round 2

Reviewer 2 Report

This paper proposed a robust hybrid observer used for tracking control. It is an interesting research topic in the robot field. The paper presented the research results in the area, which should be helpful and instructive for researchers in robot control with multiple constraints. In summary, the data of this work is detailed, and the relevant experimental parameters have been studied. This paper is well-written and organized and contains some interesting results. I recommend this work be published in Sensors after minor revision. Some concerns should be addressed before publication.

1) The Abstract should be better articulated.

2) The format of references in the dissertation is not uniform, citing fewer papers in this journal, needs to increase references in this journal.

3) Description and explanation of units should be added in Fig 6.

4) In Fig 9, the experimental results are not described and quantified in detail.

5) At present, there is a lot of work on tracking control, such as "Sub-super stochastic matrix with applications to bipartite tracking control over signed networks".  Compared with the existing work, what are the highlights of this paper?

Author Response

Response to Reviewer 2 Comments

Point 1:

This paper proposed a robust hybrid observer used for tracking control. It is an interesting research topic in the robot field. The paper presented the research results in the area, which should be helpful and instructive for researchers in robot control with multiple constraints. In summary, the data of this work is detailed, and the relevant experimental parameters have been studied. This paper is well-written and organized and contains some interesting results. I recommend this work be published in Sensors after minor revision. Some concerns should be addressed before publication.

Response 1:

Thanks for your positive comment.

Point 2:

The Abstract should be better articulated.

Response 2:

Thanks for your helpful comments. The abstract has been improved and articulated in paper.

Point 3:

The format of references in the dissertation is not uniform, citing fewer papers in this journal, needs to increase references in this journal.

Response 3:

Thanks for your comments. The format of references has been improved and some related references has been added.

Point 4:

Description and explanation of units should be added in Fig 6.

Response 4:

Thanks for your helpful comments. Description and explanation of Fig 6 has been added in paper.

Point 5:

In Fig 9, the experimental results are not described and quantified in detail.

Response 5:

Thanks for your helpful comments. The experimental results are described and quantified in detail,which has been added in paper.

Point 6: At present, there is a lot of work on tracking control, such as "Sub-super stochastic matrix with applications to bipartite tracking control over signed networks".  Compared with the existing work, what are the highlights of this paper?

Response 6:

Thanks for your comments. Compared with the existing work, what are the highlights of this paper is that: a hybrid state/disturbance observer-based multiple-constraint control mechanism to cope with the manipulator with multiple constraints. Here, we proposes a hybrid state/disturbance observer to estimate simultaneously the unmeasurable states and external disturbance. And an barrier Lyapunov function (BLF) is proposed and implemented to handle output saturation constraints, and a back-stepping control method is developed to provide sufficient control performance under multiple constraints. While the exists works don’t take the constraints into cosntraints, which leads to the poor control performance.

Again, the authors really appreciate the helpful suggestions and comments from Editor and reviewers.

Reviewer 3 Report

The paper should be supported with discussion. It should be outlined to use 2DOF robot manipulator for tracking control. 

Author Response

Response to Reviewer 3 Comments

Point 1:

The paper should be supported with discussion. It should be outlined to use 2DOF robot manipulator for tracking control.

Response 1:

Thanks for your valuable comment. The simulation on 2-DOF robot and 6-DOF robot has been carried out and added in paper. Experiment results show that the proposed method can realize high-precious tracking control on 2-DOF robot and 6-DOF robot.

Again, the authors really appreciate the helpful suggestions and comments from Editor and reviewers.